# Visual Object Detection with DETR to Support Video-Diagnosis Using Conference Tools

**Attila Biró** [1,2,3] **, Katalin Tünde Jánosi-Rancz** [4] **, László Szilágyi** [4,5] **, Antonio Ignacio Cuesta-Vargas** [2,3,6] **, Jaime Martín-Martín** [3,7] **and Sándor Miklós Szilágyi** [1,*]

[1] Department of Electrical Engineering and Information Technology, George Emil Palade University of Medicine, Pharmacy, Science, and Technology of Targu Mures, Str. Nicolae Iorga, Nr. 1, 540088 Targu Mures, Romania; abiro@uma.es

[2] Department of Physiotherapy, University of Malaga, 29071 Malaga, Spain; acuesta@uma.es

[3] Biomedical Research Institute of Malaga (IBIMA), 29590 Malaga, Spain; jaimemartinmartin@uma.es

[4] Computational Intelligence Research Group, Sapientia Hungarian University of Transylvania, 540485 Targu Mures, Romania; tsuto@ms.sapientia.ro (K.T.J.-R.); lalo@ms.sapientia.ro (L.S.)

[5] Physiological Controls Research Center, Óbuda University, 1034 Budapest, Hungary

[6] Faculty of Health Science, School of Clinical Science, Queensland University Technology, Brisbane 4000, Australia

[7] Legal and Forensic Medicine Area, Department of Human Anatomy, Legal Medicine and History of Science, Faculty of Medicine, University of Malaga, 29071 Malaga, Spain

* Correspondence: sandor.szilagyi@umfst.ro; Tel.: +40-732-131-974

**Abstract:** Real-time multilingual phrase detection from/during online video presentations—to support instant remote diagnostics—requires near real-time visual (textual) object detection and preprocessing for further analysis. Connecting remote specialists and sharing specific ideas is most effective using the native language. The main objective of this paper is to analyze and propose—through DEtection TRansformer (DETR) models, architectures, hyperparameters—recommendation, and specific procedures with simplified methods to achieve reasonable accuracy to support real-time textual object detection for further analysis. The development of real-time video conference translation based on artificial intelligence supported solutions has a relevant impact in the health sector, especially on clinical practice via better video consultation (VC) or remote diagnosis. The importance of this development was augmented by the COVID-19 pandemic. The challenge of this topic is connected to the variety of languages and dialects that the involved specialists speak and that usually needs human translator proxies which can be substituted by AI-enabled technological pipelines. The sensitivity of visual textual element localization is directly connected to complexity, quality, and the variety of collected training data sets. In this research, we investigated the DETR model with several variations. The research highlights the differences of the most prominent real-time object detectors: YOLO4, DETR, and Detectron2, and brings AI-based novelty to collaborative solutions combined with OCR. The performance of the procedures was evaluated through two research phases: a 248/512 (Phase1/Phase2) record train data set, with a 55/110 set of validated data instances for 7/10 application categories and 3/3 object categories, using the same object categories for annotation. The achieved score breaks the expected values in terms of visual text detection scope, giving high detection accuracy of textual data, the mean average precision ranging from 0.4 to 0.65.

**Keywords:** object visual detection; DETR; multilingual OCR; real-time translation; remote diagnostics; YOLO4; Detectron2; realtime text detection; assessment

## 1. Introduction

Nowadays, the importance of multilingual translation has grown exponentially thanks to the impact of the COVID-19 pandemic [1,2]. Visual Object Detection [3,4] (moreover, Textual Object Detection), as a central topic of video conference-based remote diagnosis and consultancy [5] has intensified the digital transformation in the medical field. The

issue generates a new 'realtime multilingual data processing' market to catalyze the elimination of existing communication barriers of multilingual video conferences. The adoption of Visual Object Detection (VOD) to the healthcare domain will increase the efficacy of remote diagnosis processes [2], reducing the reaction time in remote consultations. Furthermore, optical character recognition (OCR) is becoming more popular [6], since it is considered a central pillar of many advanced technologies, particularly related to real-time object detection.

The Textual and Visual Object Detection (TVOD) method analyzed popular algorythms and models in object detection (e.g., Fast R-CNN, Faster R-CNN, Region-based Convolutional Neural Networks, Region-based Fully Convolutional Network, as well as realtime object detectors such as YOLO, DETECTRON and DETR) to syntetize the outcomes and to provide reliable outcomes for a new type of collaborative working model. Our research results can be easily adapted to the Sports Safety field using video conference tools or platforms.

This research validates the feasibility of VOD [3,4] topics, using several 'realtime object detection Artificial Intelligence' (AI) models and architectures to support the non-supervised communication in Video-Diagnosis, using Conference Tools. Unfortunately, such systems, ready for multilingual interpretation support are not available to our best knowledge, and such attempts are not visible in the literature.

The outcomes of this paper are crucial in the sense that it applies a feasible approach for communication bridge to health scientists as an invisible technological layer (the 'realtime multilingual translator engine'), using an artificial intelligence-based pipeline [7], to support collaborative video conferencing solutions.

This multilingual collaboration part of research [8] has significant importance and adapts technologies (OCR, AI) from other sectors ('Information Technology' (IT), Telecom, Industry) to Healthcare. The reason, we started the research: Polysemy, ambiguity, individual usage trends, and natural changes of importance over time contribute to the fact that languages are difficult to grasp. In some specific languages, it is not enough to identify just the phrases on the screen, because—based on the context—they might have different meanings. In the case of languages such as the 22 Indian languages [9], Chinese or Japanese, it will raise out the contextualization topic. While translating a specific word, phrase, or sentence, the system considers broader additional information that clarifies the role of understanding the text in question. However, it is not obvious what could and should be considered as context, when combined with visual information and beneficial context information.

These topics generate a central issue for human translators. To get a proper interpretation while attending a dedicated technical health-based conference call, attendees must submit their presentation with all relevant information a couple of days before the live session. In the case of multilingual (ad hoc) video consultancies, where every minute matters, there is no place for such preparation.

There are existing models [10,11] and methods for bidirectional translation [12,13], but in the case of multilingual and multiuser conferences, when the involved parties are speaking in languages that are not coming from the same language families, the complexity of translation is becoming high, increasing the latency time of translation as well as decreasing the translation quality.

Challenging topics are solved currently with human interpreters, but this kind of setup is expensive and requires extra arrangements. To substitute the human factor with a highly reliable solution [13,14], several research works have been conducted [15] providing various approaches for different setups. This paper proposes a ('DEtection TRansformer') DETR [16–19] based solution design and procedure for Textual (Visual) Object Detection and will search the threshold where we can achieve feasible accuracy to support remote multilingual video diagnostics.

For this research on the validity and feasibility checking side, deep learning based model structures has been applied, such as YOLO4 ('You Only Look Once') [20–22], the

DETR ('DEtection TRansformer') [16,17,23] and the Detectron2 [24–28]. We conducted the experiments in two phases: the first round contains the technical minimum training data set for reasonable statistical learning models and analyses different model architectures' hyperparameters. Further on, we updated the data collection and annotation protocol based on these experiments. Finally, we doubled the initial training and validation data sets to improve the experiments based on the lessons learned from the first phase.

We analysed other approaches on the market (such as in the case of [17]) where the experiments were they performed on COCO 2017 detections with 118 k training images and 5 k validation images. Our scope was to find a feasible solution on minimal required training data and accessible environment for healthcare sector. Our research provides solutions where the model requires a minimum training data set [29].

The research goal of both phases was to run limited validation-type experiment rounds—on each candidate group—and to summarize their results, and simultaneously to measure, compare, and analyze the outcomes, as well as to bring specific requirements about input data and preparation process efficacy. The main scope of the experiments is to prove that the detection and localization of textual and other visual element types are possible with high accuracy (even on limited training data sets), by application of screenshot images. Furthermore, using AI models already established in other sectors, we can provide outcomes with a high degree of precision to detect the textual bounding boxes for videoconference tools. These models might be part of a further cooperative communication layer between video diagnosis [5] attendees.

The novelty and innovative content of this research is the adaptation of already known AI solutions and Machine Learning algorithms from other fields to healthcare to save time, save human resources, improve the efficacy of multilingual collaboration tools, break multilingual limits and save lives. In this way, the outcomes and results can quickly cover other use cases with minimal effort.

## 2. Materials and Methods

Unlike traditional computer vision techniques, DETR approaches object detection as a direct set prediction problem. It relies on a set-based global loss, which forces unique predictions via bipartite matching, and a "Transformer Encoder-Decoder" architecture [18].

To validate DETR's feasibility, we conducted additional experiments with three of the most promising real-time object detector models, YOLO4, Detectron2, and DETR, each with different model variations.

Given a fixed small set of learned object queries, DETR reasons about the relations of the objects and the global image context [30] to output the final set of predictions in parallel directly. Due to this parallel nature, DETR is very fast and efficient, and for this reason, we propose to employ the End-to-End Textual Object Detection with Transformers [18]. Figure 1 shows the concept of DETR.

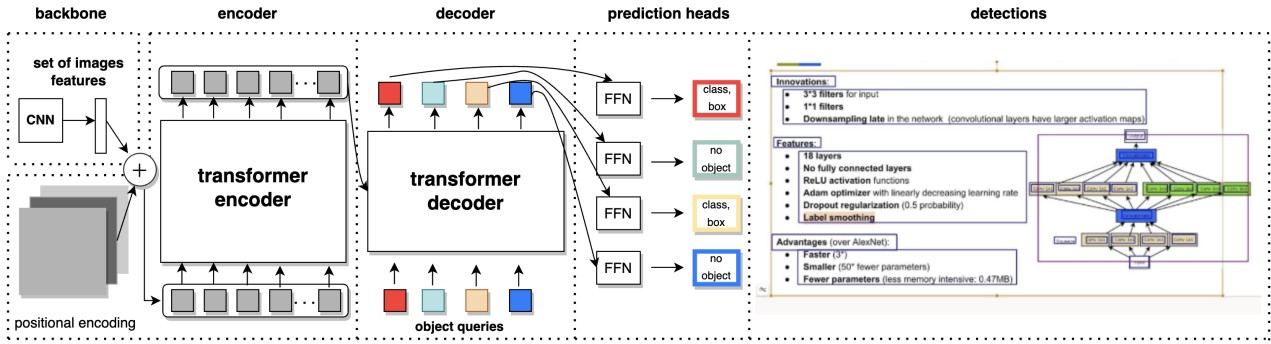

**Figure 1.** DETR architecture as presented in [17].

DETR is a PyTorch model established by Facebook based on the Transformer methodology [31], where the model's backbone is a Resnet50 image classification model, which is

fine-tuned with a Transformer based classification head for detecting relevant bounding boxes (in our case textual boxes, such as word, sentence, phrase) and assigning them to the appropriate predefined category. Transformers were introduced by Vaswani et al. [32] as a new attention-based building block for machine translation [17]. For our use case, we implemented a custom-defined classification based on four class types. The input data does not have to be the same shape (same as Detectron2). The annotation format is JSON.

### 2.1. Data

We prepared data sets from real-life use cases for our research experiments, with data taken from real video conferences and applications for this requirement. Due to the lack of appropriate options, we did not use any predefined annotated data sets. Instead, a dedicated Python picture extractor has made the data acquisition process to gain relevant potential pictures from videos. It has been used for manual data collection as well.

There were collected, filtered, and classified thousands of raw pictures. Finally, we get the data set for two independent research phases. The first one was composed of 248 train data set records, with 55 sets of validation data instances for seven application categories and three object categories, providing the necessary minimum threshold number for a specific data science project. The second phase had 512 train data set records, with 110 sets of validation data instances for ten application categories and three object categories. Human experts have made the data labeling based on a dedicated annotation and data processing protocol. The input data has been adjusted for different models (YOLO4, DETR, Detectron2) using Python scripts. Human experts performed the annotation process using the Visual Object Tagging Tool (VOTT) [33] application on Windows10 and Mac systems, as well as the Label Image Tagging Tool (LabelImg) [34] on Linux or Mac. The experiments were implemented in Python on the Google Colab Pro platform.

### 2.2. Experimental Environment

Our experiments were run on **Config 1 PC** with the following configuration: MBO Gigabyte Z390 Aorus Pro, CPU INTEL Core i7-8700K 3.7 GHz 12 MB LGA1151, DDR4 32 GB 3600 MHz Kingston HyperX Predator Black CL17 KIT2, VGA MSI RTX 2080 Ventus 8 GB, SSD M.2 SAMSUNG 970 Pro 1 TB, Corsair RMx (2018) 750 W Modular 80+ Gold) as well as on **Google Colab**- and **Google Colab Pro** (to be able to use accelerated hardware) platform.

### 2.3. Data Classification

From an algorithmic perspective, there is no particular limit for the number of object categories or instances. From the task perspective, using "Visual Textual Object Detection" as an approach, different levels of object differentiation can be valid. For our use case, we proposed four categories as follows:

1.  **Class 1** (TEXT) are the text blocks, and contains text only;
2.  **Class 2** (STATIC) contains buttons or texts and need additional dictionaries for further processing;
3.  **Class 3** (NO TRANSLATE) contains mixed elements unnecessary to process. They prevent sections from being recognized as any other category, providing data processing according to GDPR;
4.  **Class 4** (CONTAINER) is defined as inter-dependency element. They are container objects to connect larger blocks of interdependent textual elements.

### 2.4. Architecture

DETR is a PyTorch model of Facebook using the Transformer methodology (see Section 2 and Figure 1), and matches Faster R-CNN [35] with a ResNet-50 [36] than with other options we tried during our experiments. The architectue consists of three main parts: a CNN backbone for extracting compact feature representations, a simple encoder-decoder transformer, and a simple FFN for predicting the final detection [17]. The image input size was defined arbitrarily. Experiments have shown that images resized between

800 and 1333 pixels produce the best results. Initially, we chose as image size 800 pixels, then increased it to 1333 pixels to improve the accuracy. Another important fact about the architecture is that it uses Transformer attention blocks.

*2.5. Algorithm Logic*

As it has been mentioned by N. Carion et al. "DETR model directly predicts (in parallel) the final set of detections by combining a common CNN with a transformer architecture" [17]. In a generic way, the Transformer contains five major parts: (1) *Input transformation for Encoder*, (2) *Input Transformation for Decoder*, (3) *Encoder*—to extract the contextual features from input sequence through multi-head self-attention mechanism, (4) *Decoder*—from encoder output to generate the next prediction, and the (5) *Final Layer* to map decoder's output to the object probability space [19].

As is shown in Figure 1, the logic of the algorithm is structured as follows: (a) the backbone ResNet50 predicts image features from the image, (b) the algorithm calculates the spatial positional encoding for each block of the predicted image resized to the size range of 800 to 1333 pixels, (c) run Transformer module on the blocks with the belonging positional encoding, (d) each Transformer head predicts the bounding box and the object within it, and finally (e) the Transformer predicts the number of objects in the image [19].

There are two pillars of the direct set predictions in detection algorithm as shown in Figure 1: (1) a set prediction loss, which enables unique matching between predicted and ground truth boxes, as well as (2) a dedicated architecture which enables the prediction of a set of objects and models the relations in one step [17].

DETR's transformer uses positional encodings at every attention layer, as shown in Figure 2. From the CNN backbone, the image features are passed through the transformer encoder, in parallel with spatial positional encoding. These features are then added to queries and keys within the 'multi-head self-attention' layers. After that, the decoder gets the queries, output (i) positional encoding, and (ii) encoder memory. Finally there are generated the final set of 'predicted class labels' and the 'textual bounding boxes', through the 'multiple multihead self-attention' and 'decoder-encoder' attention [17].

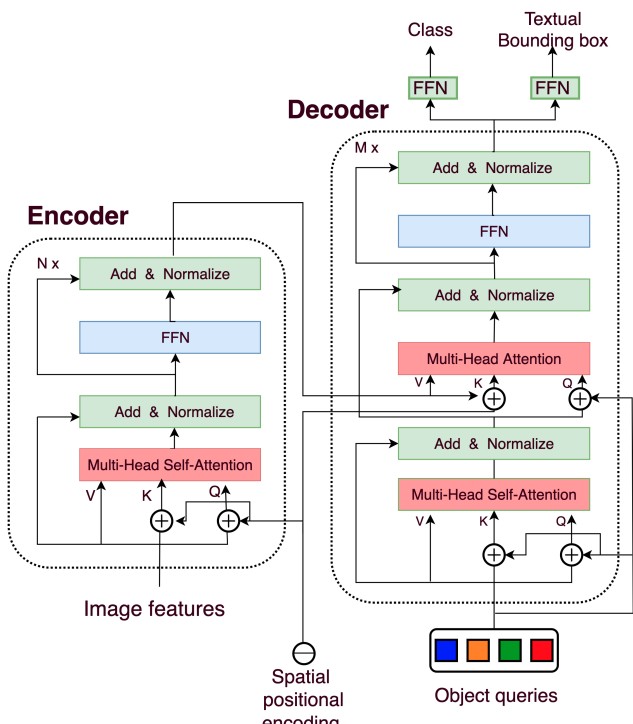

**Figure 2.** Detailed DETR transformer architecture as presented in [17].

2.5.1. Advantages

From a textual object detection perspective, DETR offers a simpler, more flexible pipeline architecture that requires fewer heuristics and casts the object detection (in our case text detection) task as an image-to-set problem [17]. Traditional computer vision models use complex pipelines with custom layers to localize objects in an image and extract the features. Conversely, DETR replaces this approach with a simpler neural network (NN), resulting in deep end-to-end learning (DL) solution to the object detection problem [17]. In TVOD research, the most important advantages are as follows: (a) Fast model—12 frames per second on GPU with the most accurate model configuration, (b) State-of-the-art accuracy as of 2020, (c) it has a straightforward implementation thanks to the PyTorch wrapper.

In our experiments we used a modest testing environment (see Section 2.2), which can be accessible by anyone currently on market.

2.5.2. Disadvantages of DETR from the Textual Object Detection Perspective

According to DETR research paper [19], one of the main disadvantages of DETR from the "textual object detection" perspective is that the detection efficacy on small targets is usually poor. In order to solve this issue, the model needs significant improvements [37]. In the case of videoconference tools, this deficiency is not present. In this research, we faced the following: (a) it requires heavy computation power (minimum 8 GB GPU memory required for the model), (b) the Transformer is a black box for interactive interoperability, (c) the Transformer network requires the largest amount of training data.

*2.6. Pre-Processing*

Nowadays, the Visual Transformers are the subject of dedicated focus because of their ability to comprehensively interpret the image's information. These model types have high fine tunability and higher accuracy than traditional image processing neural networks.

"DETR approaches object detection as a direct set prediction problem. The model uses a set-based global loss, which forces unique predictions via bipartite matching, and a Transformer encoder-decoder architecture" [30]. Each experiment and validation research (see Sections 3.1 and 3.2) in the same research phase was conducted on the same training- and validation dataset, application categories, and object categories, resulting in: (i) Phase 1: 248 images with 55 validation, 7 application categories, and 3 object categories, (ii) Phase 2: 512 annotated images, with 11 validation, 10 application categories, and 3 object categories. In addition, the prepared dataset contains manually prepared, multi-oriented scene text with quadrangle bounding box annotation.

We tested the model's limits using a minimum viable set of learned object queries (see Section 2.1) on minimalist test environment (see Section 2.2). We tested the most well-known algorithms (See Tables 1 and 2) and their **variations** (See Table 3) to get the best performance index and to prove that if we choose the optimal setup, **we will not face a well-known drawback: to run until infinity the training process**. Our goal was to run some relatively limited experiment rounds and summarize the results. We made result elaborations based on average precision (AP) and mean average precision (mAP) metrics for evaluation. Furthermore, we also inspected some labeled validation examples visually for a better human evaluation. We used the following variations:

1.　Different number of epochs: 50, 150, and 1000.
2.　Different categorizations: all 4 predefined categories, or 2 categories: text + others (static, no translate, container categories merged to one category).

**Table 1.** Detectron Phase 1 Experiments and Results.

| Model | Iterations | AP$_1$ | AP$_2$ | AP$_3$ | mAP | Total Loss |
|---|---|---|---|---|---|---|
| faster_rcnn_R_50_C4_3x | 500 | 0.07034 | 0.07367 | 0.09235 | 0.06877 | 1.320 |
| faster_rcnn_R_50_DC5_3x | 1000 | 0.17659 | 0.23342 | 0.20921 | 0.17706 | 1.253 |
| faster_rcnn_R_50_FPN_3x | 700 | 0.30985 | 0.44342 | 0.36016 | 0.34255 | 1.154 |
| faster_rcnn_R_50_FPN_3x | 1000 | 0.30560 | 0.42302 | 0.34745 | 0.34160 | 1.183 |
| faster_rcnn_R_101_FPN_3x | 1000 | 0.29764 | 0.47187 | 0.43785 | 0.39341 | 1.129 |
| faster_rcnn_X_101_32x8_FPN_3x | 3000 | 0.42799 | 0.56096 | 0.56331 | 0.49369 | 0.458 |
| faster_rcnn_X_101_32x8_FPN_3x | 10,000 | 0.45223 | 0.57821 | 0.55734 | 0.48758 | 0.343 |
| retinanet_R_50_FPN_1x | 1000 | 0.31027 | 0.45998 | 0.47357 | 0.40872 | 0.495 |
| retinanet_R_50_FPN_3x | 1000 | 0.32150 | 0.46114 | 0.46953 | 0.41103 | 0.477 |
| retinanet_R_101_FPN_3x | 3000 | 0.36923 | 0.55230 | 0.55267 | 0.49589 | 0.250 |
| retinanet_R_101_FPN_3x | 10,000 | 0.37968 | 0.55938 | 0.57270 | 0.50513 | 0.189 |
| rpn_R_50_C4_1x | 1000 | | | | | 0.320 |
| rpn_R_50_FPN_1x | 3000 | | | | | 0.180 |

**Table 2.** Detectron Phase 2 Experiments and Results.

| Model | Total Loss | mAP | Workers | BATCH/IMG | IMS/BATCH | Iterations |
|---|---|---|---|---|---|---|
| faster_rcnn_R_50_C4_1x | 0.963 | 0.29753 | 4 | 128 | 6 | 1000 |
| faster_rcnn_R_50_DC5_1x | 0.967 | 0.31446 | 4 | 128 | 6 | 1000 |
| faster_rcnn_R_50_FPN_1x | 0.912 | 0.33348 | 4 | 128 | 6 | 1000 |
| faster_rcnn_R_50_C4_3x | 0.912 | 0.34338 | 4 | 128 | 6 | 1000 |
| faster_rcnn_R_50_DC5_3x | 0.886 | 0.34033 | 4 | 128 | 6 | 1000 |
| faster_rcnn_R_50_FPN_3x | 0.912 | 0.36219 | 4 | 128 | 6 | 1000 |
| faster_rcnn_R_50_FPN_3x | 0.707 | 0.39064 | 4 | 128 | 6 | 3000 |
| faster_rcnn_R_101_C4_3x | 0.857 | 0.32096 | 4 | 128 | 6 | 1000 |
| faster_rcnn_R_101_DC5_3x | 0.882 | 0.30924 | 4 | 128 | 6 | 1000 |
| faster_rcnn_R_101_FPN_3x | 0.869 | 0.32056 | 4 | 128 | 6 | 1000 |
| faster_rcnn_X_101_32x8_FPN_3x | 0.998 | 0.30455 | 2 | 64 | 3 | 1000 |
| retinanet_R_50_FPN_1x | 0.661 | 0.24791 | 4 | 128 | 6 | 1000 |
| retinanet_R_50_FPN_3x | 0.568 | 0.25690 | 4 | 128 | 6 | 1000 |
| retinanet_R_101_FPN_3x | 0.564 | 0.28165 | 4 | 128 | 6 | 1000 |
| rpn_R_50_C4_1x | 0.350 | 0.24046 | 4 | 128 | 6 | 1000 |
| rpn_R_50_FPN_1x | 0.376 | 0.35200 | 4 | 128 | 6 | 1000 |

**Table 3.** YOLO4 experiments and results.

| Epochs | YOLO4 TINY 640 (16) | | | | YOLO4 TINY 416 (32) | | | | YOLO4 TINY 800 (16) | | | |
|---|---|---|---|---|---|---|---|---|---|---|---|---|
| | AP$_1$ | AP$_2$ | AP$_3$ | mAP | AP$_1$ | AP$_2$ | AP$_3$ | mAP | AP$_1$ | AP$_2$ | AP$_3$ | mAP |
| 10 | | | | | 0.029 | 0.040 | 0.130 | 0.066 | 0.145 | 0.199 | 0.398 | 0.247 |
| 20 | 0.164 | 0.153 | 0.352 | 0.223 | 0.051 | 0.039 | 0.117 | 0.069 | 0.172 | 0.141 | 0.312 | 0.208 |
| 30 | 0.158 | 0.136 | 0.352 | 0.215 | 0.069 | 0.047 | 0.131 | 0.082 | 0.280 | 0.344 | 0.291 | 0.305 |
| 40 | 0.272 | 0.150 | 0.343 | 0.255 | 0.064 | 0.051 | 0.127 | 0.081 | 0.281 | 0.265 | 0.392 | 0.312 |
| 50 | 0.255 | 0.156 | 0.372 | 0.261 | 0.070 | 0.063 | 0.118 | 0.084 | 0.257 | 0.332 | 0.445 | 0.345 |
| 60 | 0.270 | 0.179 | 0.337 | 0.262 | 0.068 | 0.065 | 0.113 | 0.084 | 0.269 | 0.222 | 0.472 | 0.321 |
| 70 | 0.308 | 0.129 | 0.339 | 0.259 | 0.078 | 0.058 | 0.131 | 0.089 | 0.290 | 0.400 | 0.484 | 0.392 |
| 80 | 0.296 | 0.105 | 0.356 | 0.252 | 0.087 | 0.062 | 0.130 | 0.093 | 0.326 | 0.359 | 0.485 | 0.390 |
| 90 | 0.343 | 0.231 | 0.364 | 0.313 | 0.080 | 0.059 | 0.138 | 0.092 | 0.347 | 0.405 | 0.485 | **0.412** |
| 100 | 0.350 | 0.152 | 0.418 | 0.307 | 0.087 | 0.069 | 0.133 | 0.096 | 0.339 | 0.379 | 0.481 | 0.400 |
| 110 | 0.326 | 0.180 | 0.408 | 0.305 | 0.090 | 0.069 | 0.137 | 0.099 | 0.335 | 0.395 | 0.486 | 0.405 |
| 120 | 0.326 | 0.236 | 0.393 | **0.348** | 0.092 | 0.072 | 0.137 | 0.100 | 0.355 | 0.399 | 0.479 | 0.411 |
| 130 | 0.300 | 0.175 | 0.410 | 0.295 | | | | | 0.343 | 0.411 | 0.486 | **0.413** |
| 140 | nan | nan | nan | nan | | | | | 0.340 | 0.411 | 0.483 | 0.411 |
| 150 | | | | | | | | | 0.345 | 0.409 | 0.483 | 0.412 |

**Table 3.** *Cont.*

| Epochs | YOLO4 LARGE 480 (16) | | | | YOLO4 TINY 416 (16) | | | | YOLO4 TINY 800 (16, 0.005) | | | |
|---|---|---|---|---|---|---|---|---|---|---|---|---|
| | $AP_1$ | $AP_2$ | $AP_3$ | mAP | $AP_1$ | $AP_2$ | $AP_3$ | mAP | $AP_1$ | $AP_2$ | $AP_3$ | mAP |
| 10 | 0.140 | 0.023 | 0.190 | 0.118 | 0.035 | 0.008 | 0.104 | 0.049 | 0.315 | 0.337 | 0.451 | 0.368 |
| 20 | 0.206 | 0.061 | 0.263 | 0.177 | 0.049 | 0.026 | 0.093 | 0.056 | 0.373 | 0.271 | 0.432 | 0.359 |
| 30 | 0.214 | 0.024 | 0.266 | 0.168 | 0.043 | 0.030 | 0.109 | 0.061 | 0.440 | 0.247 | 0.372 | 0.353 |
| 40 | 0.247 | 0.041 | 0.288 | 0.192 | 0.064 | 0.015 | 0.102 | 0.060 | nan | nan | nan | nan |
| 50 | 0.228 | 0.024 | 0.300 | 0.184 | 0.062 | 0.031 | 0.111 | 0.068 | | | | |
| 60 | 0.268 | 0.057 | 0.371 | 0.232 | 0.049 | 0.028 | 0.118 | 0.065 | | | | |
| 70 | 0.252 | 0.042 | 0.344 | 0.213 | 0.061 | 0.031 | 0.105 | 0.065 | | | | |
| 80 | 0.272 | 0.057 | 0.323 | 0.217 | 0.076 | 0.027 | 0.120 | 0.074 | | | | |
| 90 | 0.258 | 0.012 | 0.283 | 0.184 | 0.079 | 0.018 | 0.142 | 0.080 | | | | |
| 100 | 0.281 | 0.040 | 0.316 | 0.212 | 0.084 | 0.029 | 0.110 | 0.074 | | | | |
| 110 | 0.235 | 0.011 | 0.298 | 0.181 | 0.078 | 0.034 | 0.132 | 0.081 | | | | |
| 120 | 0.262 | 0.034 | 0.299 | 0.198 | 0.080 | 0.027 | 0.140 | 0.082 | | | | |
| 130 | | | | | 0.087 | 0.023 | 0.130 | 0.080 | | | | |
| 140 | | | | | 0.086 | 0.022 | 0.135 | 0.081 | | | | |
| 150 | | | | | 0.087 | 0.024 | 0.133 | 0.081 | | | | |

### 2.7. Decision Making

Based on the initial training results, we recognized two bottlenecks to be managed: (a) the results do not meet the expected thresholds, and (b) the training still has some reserves. These are visible in Figures 3 and 4.

Based on our findings, we redesigned the system to produce results despite the training data set limitation. Experimental findings showed that categorizing four classes for textual object detection requires much more training data than is proposed by standards. A broader complexity of options and outcomes will not be enough to support a feasible multilingual VC consultancy solution. Therefore, we modified the initial training data set by merging three categories into one, yielding the following model with the results shown in Figure 5, which can be compared with the non-clustered solution (see Figure 6). Following are the changes to the classes:

- **New Class 1**, which corresponds to the previous Class 1 and requires translation,
- **New Class 2**, which corresponds to the union of previous Classes 2, 3, and 4, and requires no translation.

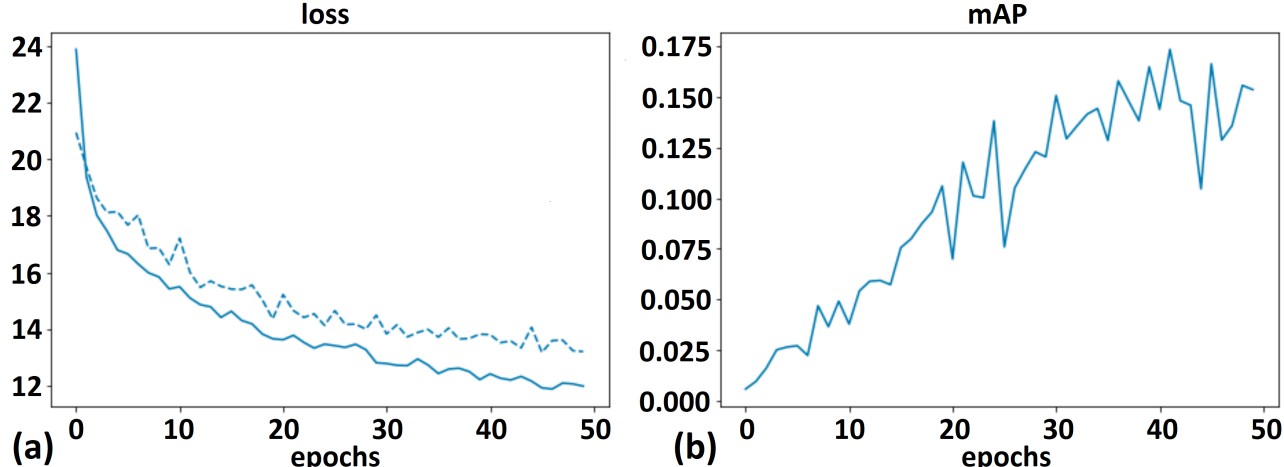

**Figure 3.** DETR results: (**a**) loss function obtained for the training set (solid line) and the validation set (dashed line) during 50 epochs, for all categories (1st experiment); (**b**) mAP scores obtained for the validation set.

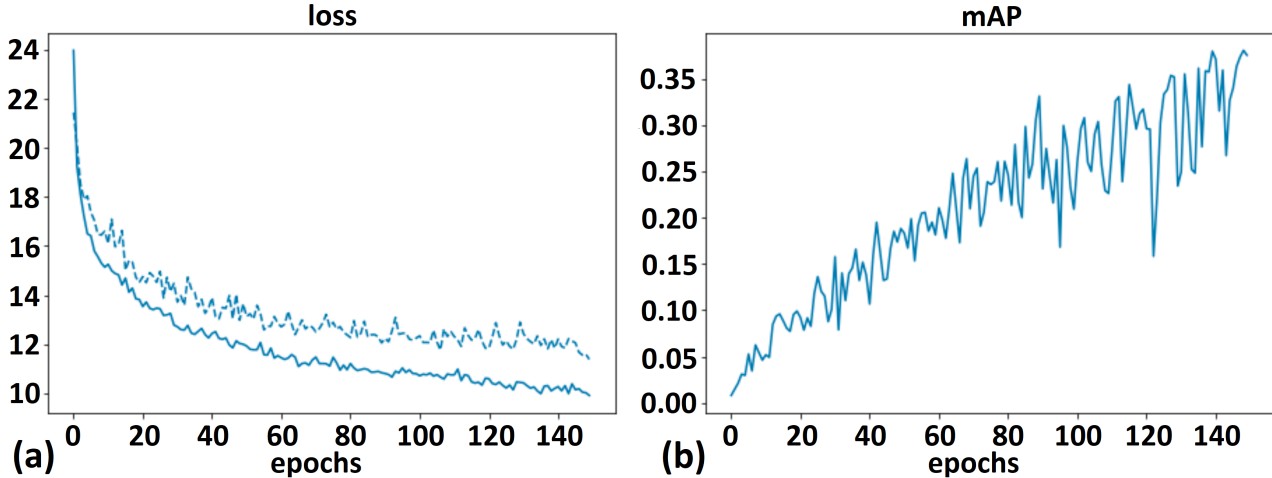

**Figure 4.** DETR results: (**a**) loss function obtained for the training set (solid line) and the validation set (dashed line) during 150 epochs, for all categories (2nd experiment); (**b**) mAP scores obtained for the validation set.

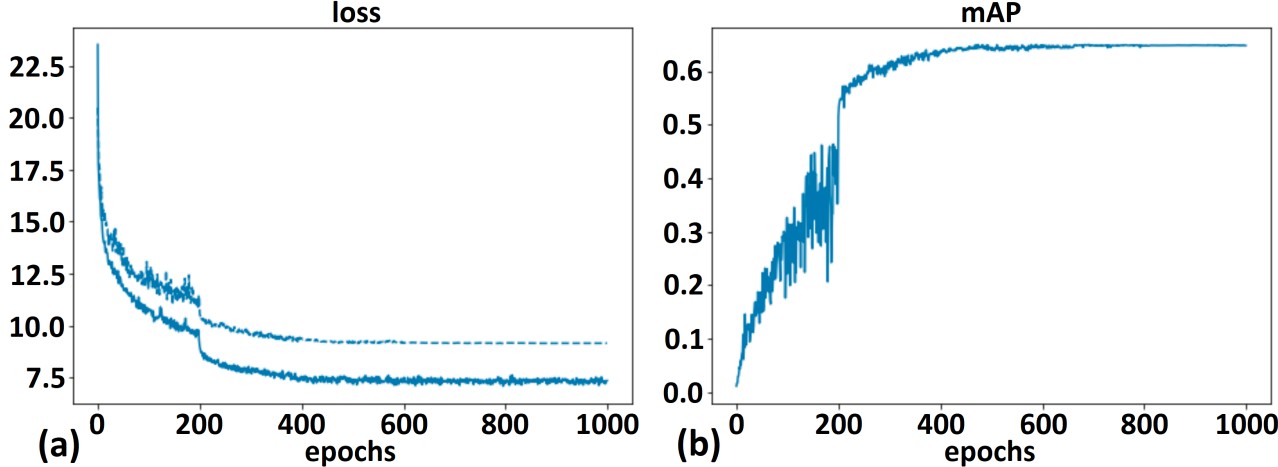

**Figure 5.** DETR results: (**a**) loss function obtained for the training set (solid line) and the validation set (dashed line) during 1000 epochs, for two categories (4th experiment); (**b**) mAP scores obtained for the validation set.

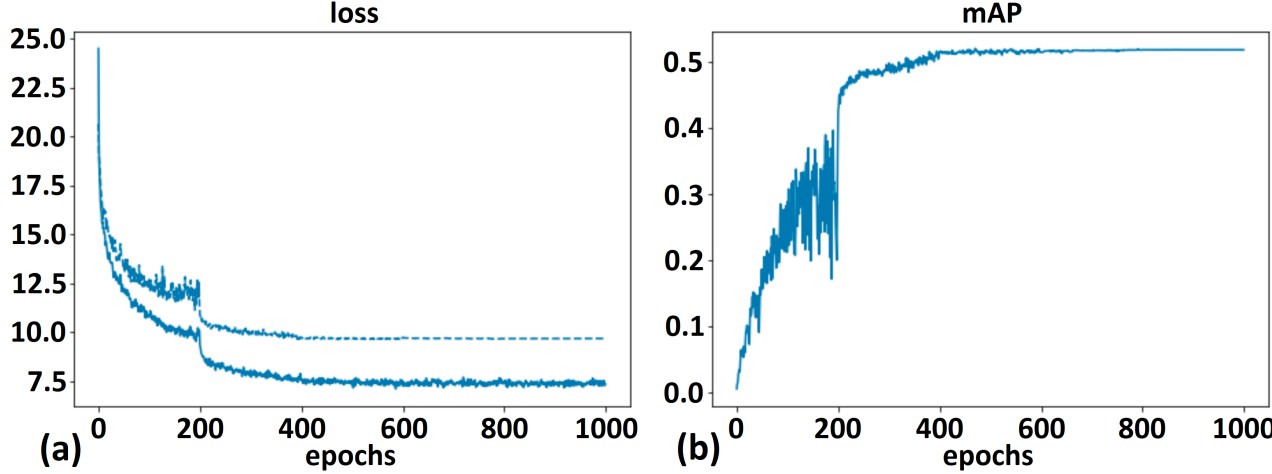

**Figure 6.** DETR results: (**a**) loss function obtained for the training set (solid line) and the validation set (dashed line) during 1000 epochs, for all categories (3rd experiment); (**b**) mAP scores obtained for the validation set.

## 3. Data Analysis

For evaluation, we used average precision (AP) and mean average precision (mAP) [38], computed as follows

$$\text{mAP} = \frac{1}{n} \sum_{k=1}^{n} \text{AP}_k \, , \tag{1}$$

where $\text{AP}_k$ represents the average precision of class $k$, and $n$ is the number of classes.

### 3.1. Alternative No. 1: YOLO4 Model and Analysis

YOLO offers a wide array of model variations, versions, architecture types, implementations, and underlying AI frameworks. Figure 7 shows the YOLO4 architectures, with *Input* (like Images, Patches or Image Pyramid), *Backbone* (it could be: VGG16, ResNet-50, SpineNet, EfficientNet, ResNeXt-101, Darknet53, DenseNet, SqueezeNet [4], MobileNet, ShuffleNet), Neck (e.g., FPN, PANet, Bi-FPN, etc.), *Head* which could have *Dense Prediction* (RPN, YOLO, SSD, RetinaNet, FCOS, CornerNet, CenterNet, MatrixNet, etc.) in the case of One-Stage type Detector as well as *Sparse Prediction* (as Faster R-CNN, R-FCN, Mask R-CNN, Libra R-CNN, etc.) in the case of Two-Stage Detector [39].

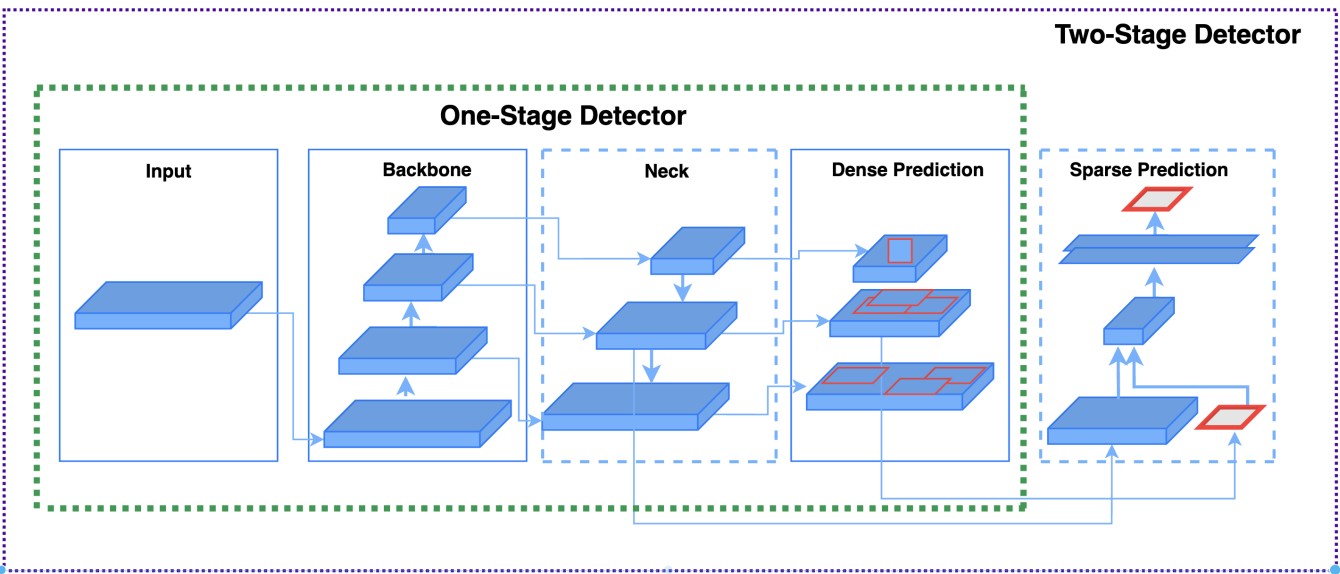

**Figure 7.** Yolo4 Architecture [39].

We started the training from randomly initialized weights or some weight sets pre-trained from scratch. This research focused on YOLO v4, using Python and TensorFlow. Data annotation has been done by VOTT [33] and LabelImg [34]. Later ones always lead to better convergence. YOLOv4 has two basic variations: standard (large [257 MB]) and tiny (small [23 MB]). These were involved in evaluation with the following configurations over 120–150 epochs: (i) Different input image sizes (between 416 × 416 and 800 × 800 pixels), (ii) Different batch sizes (8–32), (ii) Different learning rates (0.005–0.02), (iii) Different optimizers (SGD, Adam, Nadam).

### 3.2. Alternative No. 2: Detectron2 Model and Analysis

Detectron2 is a PyTorch based framework by Facebook that implements several of the widespread object detection/segmentation algorithms, such as Mask R-CNN, RetinaNet, Faster R-CNN, RPN, Fast R-CNN etc. As the meta-architecture shown in the Figure 8, we can distinguish three blocks: (1) *Backbone Network* to extract the feature maps from the input image, (2) the *Region Proposal Network* which detects the object regions from features and the (3) *Box Head* which is one of the subclasses of ROI heads [40].

In our study, only algorithms with model zoo [41] were involved in object detection. Input data were not in the same shape, unlike in the case of YOLO. Therefore, we transformed the annotations into COCO JSON format. We tried out and evaluated all proposed (best practices) relevant algorithms with the following variations: (i) different number of iterations (1000 to 10,000), (ii) different number of images per batch (4–16), (iii) different batch size per image (8–128), (iv) different warm-up iteration count (50 to 150), (v) different learning rates (0.001–0.02).

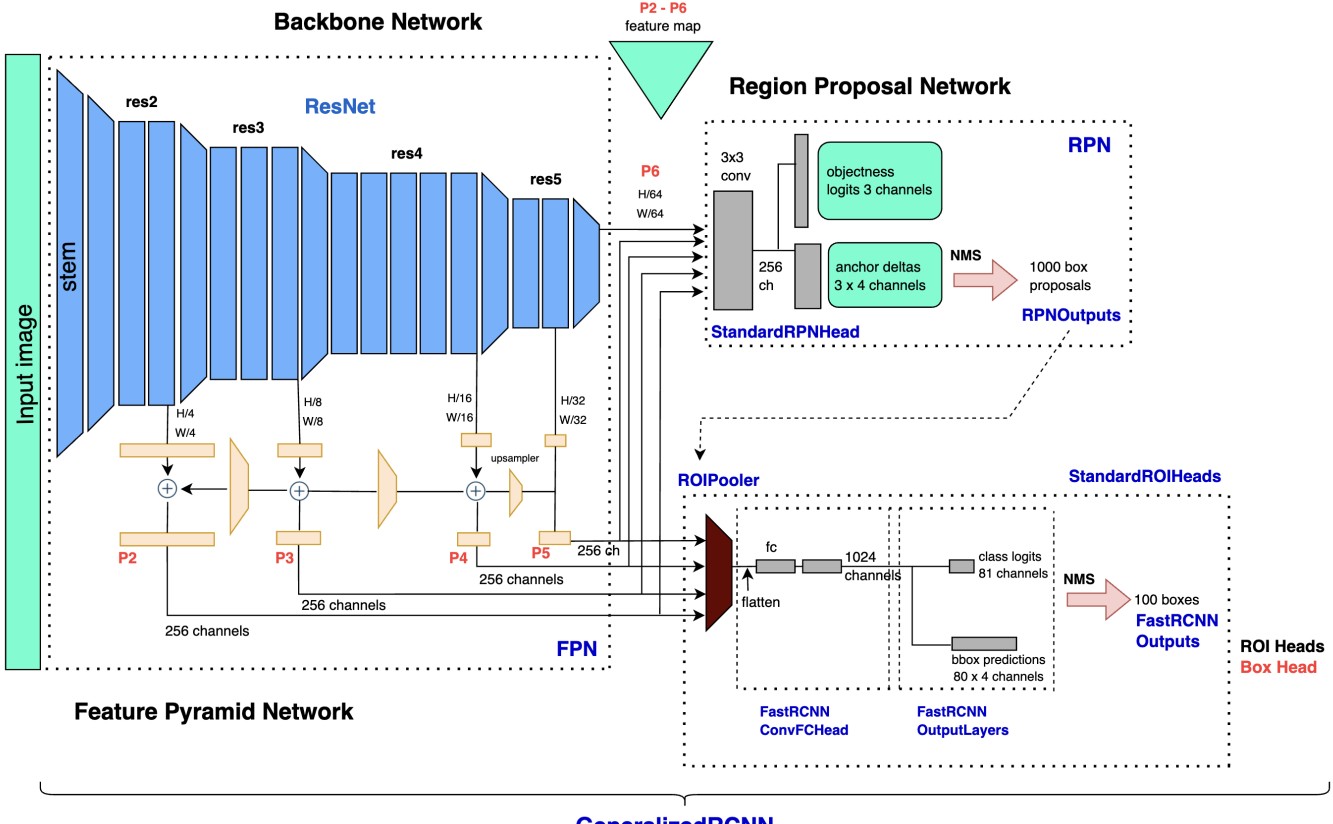

**Figure 8.** Detectron—Detailed Architecture of Base-RCNN-FPN [40].

## 4. Results

### 4.1. DETR Models

For the DETR experiment, the minimum required VRAM is 6 GB. However, the minimum VRAM is a minimum standard requirement for efficient training. One of the advantages of this model is that it could not overfit with prepared data set (see Section 2.1 for details). At approximately 500 epochs, all experiments reached their peak performance (see Figures 5 and 6). The global loss values are a combination of bounding box detection accuracy and categorization accuracy. DETR had a reasonable training time: around 9.5 h for 1000 epochs (for environment see Section 2.2) on the available training data set. Based on our experiments, the number of categories affects the model's final performance, and we can reach the top performance by using a clustering approach (synergizing the relevant classes). The best results with four categories were gained in mAP > 0.5 (see Figure 6). Nevertheless, we reached the best overall research outcome (mAP = 0.65) using a limited number of categories by clustering (see Figure 5); as a result, we obtained a better mAP score in almost the same number of epochs.

The results of the experiments were analyzed and evaluated through the values of the *loss function* for the training dataset (solid line) and the validation dataset (dashed line) as well as *mean average precision scores* for the validation dataset (solid line). In the results figures (see. Figures 3–6), the left graph shows the loss function and the right one the mAP

score, both plotted against the epoch number. Figures 3, 4 and 6 show the results of the first, second, and third experiment, with 50, 150, and 1000 epochs, respectively, using all four predefined categories—text, static, no translate, container—and DETR specific N/A categories for disputable results.

Figure 5 shows the 4th experiment results with 1000 epochs using the clustered approach. Subsequently, we merged three of the original categories into one, which resulted two new categories: (i) 'New Class 1' with original Class 1 (text) and (ii) 'New Class 2' for 'no translation object holder' composed from Class 2 (static) with Class 3 (no translate) with Class 4 (container) from original mode. Finally, Figure 9 shows the OCR output connected to the 4th experiment and confirms that the algorithm is feasible even with mAP = 0.65.

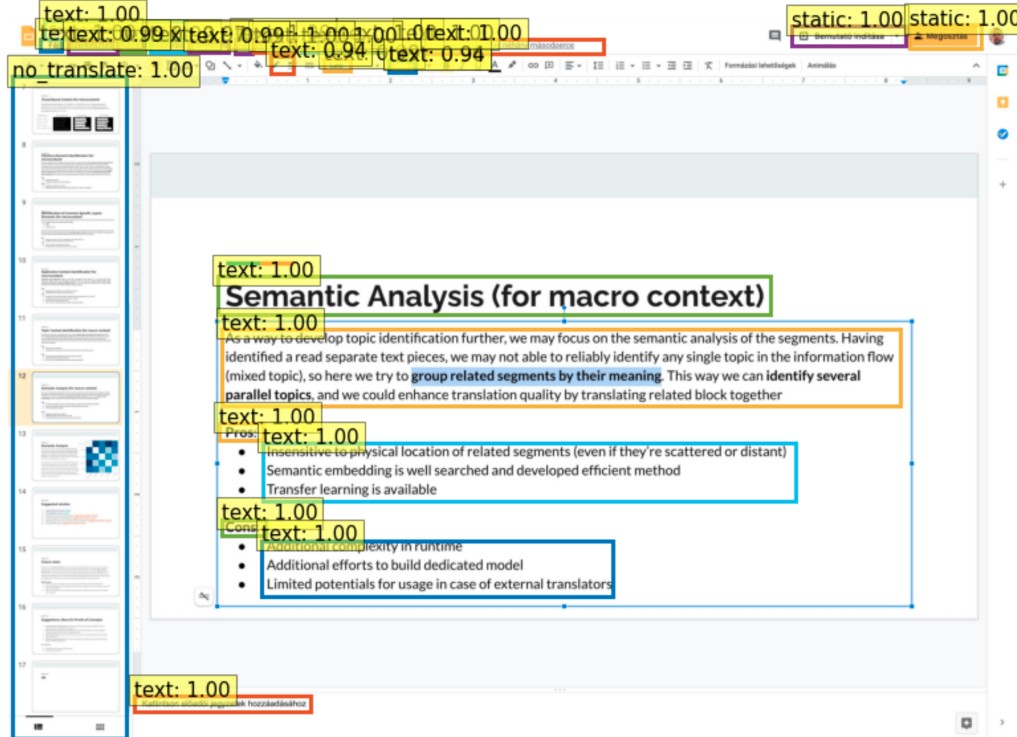

**Figure 9.** DETR: OCR Output.

*4.2. Alternative No. 1: YOLO4 Models*

It is visible in Table 3 that YOLOv4 (tiny) reached mAP values up to 0.41. We used the following settings: the pre-trained initial weights, 800 × 800 input shape, 120 epochs, SGD optimizer, initial learning rate set to 0.01, and a fixed scheduled rate decay during the run epochs. Based on the experiments we formulated the following remarks:

1.  The performance does not strictly correlate with training time (epochs).
2.  The performance almost always plateaued before terminating the learning process.
3.  Accuracy does not always correlate with the overall mAP and class-specific AP values.
4.  In comparison with other experiment groups such as Detectron2 or DETR, YOLO requires more training time (around 3–6 min per epoch), which makes the hyperparameter tuning process very slow.
5.  A very long training–over 150 epochs–becomes uncertain due to the vanishing gradient phenomenon. At this moment, the training is interrupted irrespective of the original expected length of the experiment. 'YOLO large' and 'YOLO tiny' have the same performance, 'YOLO tiny' actually tends to give better results. We realized that there is no reason to choose YOLO large for further experiments. Larger input image size provided better results than expected (up to the examined 800 × 800, at least).

6. With batch size up to 8 the relevant models were unable to learn, so we had to set the batch size to 16 as the minimum value, while 32 was the maximum value we employed due to GPU memory constraints (see Section 2.2).

According to our investigation, the optimizer type does not seem to have a significant impact in terms of performance. While advanced ones (such as Adam, Nadam) are initially faster, they tend to have the same overall accuracy as vanilla Stochastic Gradient Descent in the long run. However, Adam and Nadam apparently lead to vanishing or exploding gradients more often.

### 4.3. Alternative No. 2: Detectron2 Models

In our experimental environment described in Section 2.2, 1000 iterations last around 20–30 min, and the "iterations" here cannot correspond to YOLO epochs. The more advanced model types have higher memory requirements. In the case of Detectron2, the warm-up iteration number does not affect the outcome. The optimal learning rate is around 0.005–0.01, where the larger ones tend to explode gradients. In the case of a more advanced model, this type provides the most accurate outcome. According to our research results, performance is not strictly correlated with training time (epochs): performance almost always plateaued before the learning process was terminated. According to the experiments, as indicated in Table 1, RetinaNet and Faster-RCNN detectron models provided the best results, reaching around 0.5 in mAP.

### 4.4. Detectron Phase 2 Research

Detectron2 benefits from models that have much more rapid training than YOLO. Using the same model architecture hyperparameters involved in the earlier experiments, we renewed the research by doing the same experiments and refining our model architecture hyperparameters. However, a more suitable image processing algorithm was applied on the double amount of training and validation images, moreover improving the annotation protocol and the review process. The outcome of connected experiments gives a much better loss value at the same mAP in the case of original classes. The second phase resulted better accuracy by better training data adjustment. Table 2 shows the results of Detectron2 obtained in Phase 2. The result is quite impressive, even with four object categories. In this way, we achieved the goal of this paper, having the optimal minimal set of training and validation data, and the proposed feasible models for a multilingual use case.

### 5. Discussion

The achieved overall score breaks the expected values in a specific OCR scope, resulting in a high textual data detection accuracy with mAP = 0.65.

As it is visible on the charts of Figures 3 and 4, during first experiment with DETR (mAP = 0.174) and second experiment with DETR (mAP = 0.39), there is room for improvement by increasing the number of epochs since the model did not yet produce overfitting. During the third experiment with DETR (mAP = 0.5) depicted in Figure 6, we can see that the model reaches its best after 400 epochs, and there is only a tiny increment after that point. The model is not overfitting, but it cannot improve its performance. We observed very similar results in the fourth experiment with DETR (mAP = 0.65) exhibited in Figure 5, but with a limited number of categories. However, a considerably higher mAP score has been achieved almost in the same number of epochs. We checked the results for our Textual Object Detection at this point of research. The system had not reached the 0.90 value with mAP, just 0.65 (see Figure 9); nonetheless, the focused objects were identified accurately.

In Phase 1 of this research, we applied the DETR DL-based model structure. It used only 248 manually collected and annotated training examples and 55 validation examples to fit the technical fminimum viable model for practical statistical learning. To achieve the theoretical maximum with mAP exceeding 0.90 (e.g., in the case of context-sensitive

language detection), the model requires a minimum set of 1500 training with 200 validation data, 1700 instances combined, according to our calculations based on research experiments.

The original goal of the research was to prove that detecting textual elements with Visual Transformers and fully customizable ML models is possible with suitable accuracy by applying screenshot images.

Figure 3 shows that there is room for refinement by enriching the number of epochs because the model is not overfitted yet. Overfitting can be determined from loss functions: we can reach the starting point of overfit when the training loss decreases, but the validation loss starts to increase. Figure 4 shows the 2nd experiment results, and it is observable that we can still boost the epoch number. Figure 6 shows the 3rd experiment results. The model is reaching its best after 400 epochs, and there is only a small increment after that point. We have to change the type of experiment if we want to improve the model. The model is not overfitted yet, but it cannot improve anymore via learning. At this point, we obtained mAP = 0.5. Figure 5 shows the 4th experiment results with 1000 epochs with a clustered approach, and this experiment brings the real breakthrough; a much better score (mAP = 0.65) was reached in almost the same number of epochs (see Figure 5). Figure 9 shows the OCR output connected to the 4th experiment.

The feasibility checking and validation experiment emphasize that the models recognize text boxes and categories with high accuracy, specific to the requirements of the textual topic approach. Each experiment group gave the same level of performance (mAP between 0.4 and 0.65) irrespective of the actual model types. The main bottleneck for further improvement is the limited data quantity and quality and the coverage of all use cases. For this reason, we improved the annotation protocols, annotation review process, and image acquisition and selection guidelines to start the second research phase.

**YOLO** reached its best (mAP = 0.413) with YOLOv4 Tiny (see Table 3). YOLO's disadvantage was the slow training process. **Detectron**'s best score of mAP = 0.5 is coming from Retinanet R101 FPN 3x (see Table 1). Detectron2's disadvantage is the complex framework. Finally, **DETR**'s best of mAP = 0.65 came from DETR RESNET50 (see Figure 5). Disadvantages of this model are the slow training, separated detection, and classification model. As a further investigation, it will be compared with the results of the most promising architectures. According to the results, it can be provided a new solution to support the multilingual, collaborative distant diagnostics, breaking the language barriers with AI-supported solutions. The settings used by the best models are as follows:

1. **YOLO4 Tiny**: $800 \times 800$ input size, 120 epochs, SGD optimizer, learning rate 0.01, batch size 16;
2. **Detectron2 RetinaNet R 101 FPN 3x**: 10,000 epochs, learning rate 0.01, 6 images per batch, 128 batch per images, 150 warm-ups;
3. **DETR RESNET50**: variable input size, 500 epochs, learning rate 0.0001, batch size 2, LR decay $0.0001\times$ every 200 epochs.

During the second phase experiments, we obtained almost the same level of mAP but with much better loss values (see Table 2). Suppose we apply the clustering approach to our data classification, in other words: instead of four different classes (text, static, no translate, container), we focus just on textual objects—then our models reach the best performance on a moderate number of training data with validation and in this way we can adapt real-time object detection models to real-time consultations [5,8] and multilingual collaborations tools and solutions. From the investigated models, DETECTRON and DETR can serve best as these approaches because they will provide the same level of accuracy, but the best results will provide DETR.

*5.1. DETR Insights on Textual Object Detection*

The most important outcomes from the experiments can be summarized as follows:

1. About 6 GB of VRAM is required (at least) for efficient training;
2. The model could not be over-fitted during the experiments, and this is promising;

3. At approximately 500 epochs, all experiments reached their peak performance;
4. The global loss values are a combination of bounding box detection accuracy and categorization accuracy;
5. Moderate training time (9.5 h/1000 epochs with the available training data);
6. The number of categories affects the model's final performance—object detection with clustering [42] was a line of research connected to our scope.

### 5.2. AI for OCR and Multilingual Translations for Video Conference Consultancies

Object detection with transformers (DETR) reaches competitive performance with Faster R-CNN via a transformer encoder-decoder architecture [43]. Our experiment shows that even with a low number of pictures and moderate performance (mAP = 0.65), we can build models that recognize text box elements, e.g., phrases for collaborative work. Multilingual real-time neural translators can easily deploy this approach as an interface for collaborative and interactive translation in real-time. As a consequence of our study, to fit the model to a wide variety of languages, we can define precise requirements for a highly accurate model, together with annotation guidelines and preparation policies.

## 6. Conclusions

The paper proposed DETR for visual, textual data recognition to support future translation of multiuser videoconferences and to boost the efficacy of distance diagnosis. The Textual and VOD project analyzed different scenarios to provide reliable outcomes for a new collaborative working model. Results will be applied to various sectors (e.g., health, sports) that use remote consultations or e-collaboration tools and videoconference platforms (like Zoom, Google Meet, Microsoft Teams, Skype, Google Hangouts, GoToMeeting, Cisco Webex Meetings, Bluejeans, Intermedia Anymeeting, RingCentral Video, Zoho Meeting, Dialpad Meetings, TrueConf Online, Lifesize Go or even Youtube Live, Facebook Live, Slack Video Calls, etc.). The research validated, the feasibility of the Textual Object Detection topic, using several different models and architectures to support video diagnosis using conference tools.

The main goal of the investigation was to run some limited validation experiment rounds on each candidate group, summarize their results, compare the outcomes, and bring specific requirements about the input data and the preparation process, and the efficacy of the new approach.

In the case of multilingual conferences, to support further data processing, such as neural multilingual translation, a feasible visual (textual) object detection model can be prepared by a modest number of specific training datasets using the DETR model. When it comes to adjusting the efficacy level, it depends on clustering or object classes, quality- and quantity of training data. The solution provides satisfactory results, starting from 0.50 mAP for this use case, which is an acceptable outcome connected to data science and DL projects. Using DETR with the clustering approach for textual object detection, the prediction of bounding boxes give high accuracy at approximately 500 epochs (see Figures 5 and 6), where all experiments reach their peak performance. The results of mAP = 0.65 represent high accuracy in textual object detection, as shown in Figure 9, being promising from the collaborative, multilingual solutions perspective. It could bring a new line of development for specific applications. In VOD projects, we usually expect mAP values above 0.9 to prove the feasibility of different use cases. However, if we are only looking for texts and phrases, the expected criteria are already met above 0.5 mAP. Experiments have demonstrated that using the DETR model for a restricted number of training datasets with real-time collaboration tools is possible to strengthen multilingual collaboration with multilingual translation. The textual data object detection is feasible having a mean average precision of 0.65.

**Author Contributions:** Conceptualization, A.B., S.M.S. and A.I.C.-V.; methodology, A.B., A.I.C.-V., J.M.-M. and S.M.S.; software, A.B. and K.T.J.-R.; validation, A.B., K.T.J.-R., L.S. and S.M.S.; formal analysis, A.B., J.M.-M. and S.M.S.; investigation, A.B. and K.T.J.-R.; resources, A.B. and S.M.S.; data curation, A.B. and K.T.J.-R.; writing—original draft preparation, A.B.; writing—review and editing, A.B. and L.S.; visualization, A.B. and L.S.; supervision, A.I.C.-V., J.M.-M. and S.M.S.; project administration, A.B.; funding acquisition, A.B. and L.S. All authors have read and agreed to the published version of the manuscript.

**Funding:** This research was supported by ITware, Hungary. The work of K.T. Jánosi-Rancz and L. Szilágyi was supported by Sapientia Foundation—Institute for Scientific Research.

**Institutional Review Board Statement:** Not applicable

**Informed Consent Statement:** Not applicable.

**Data Availability Statement:** Data are available upon request. The Figures (DETR architecture, Detailed DETR transformer architecture, YOLO4 architecture, Detectron detailed architecture) and the results charts of this study (Figures 3–6, DETR OCR output) can be found in: https://doi.org/10.6084/m9.figshare.19699993 (accessed on 31 May 2022).

**Conflicts of Interest:** The authors declare no conflict of interest.

## Abbreviations

The following abbreviations are used in this manuscript:

| | |
|---|---|
| Adam | an alternative optimization algorithm |
| AI | artificial intelligence |
| AP | average precision |
| Bi-FPN | weighted bi-directional feature pyramid network |
| CenterNet | machine learning model for Anchorless Object Detection |
| CNN | convolutional neural network |
| COCO | common objects in context |
| CornerNet | new approach to object detection |
| COVID-19 | coronavirus disease |
| CPU | central processing unit |
| Darknet53 | is a convolutional neural network architecture |
| DDR | Document Domain Randomization |
| DETR | Detection Transformer |
| DenseNet | a type of convolutional neural network architecture |
| EfficientNet | a convolutional neural network architecture |
| FCOS | fully convolutional one-stage object detection |
| FPN | feature pyramid network |
| GB | gigabyte |
| GDPR | General Data Protection Regulation |
| IT | information technology |
| JSON | JavaScript object notation |
| mAP | mean average precision |
| MatrixNet | a proprietary machine learning algorithm |
| MobileNet | a convolutional neural network architecture |
| Nadam | Nesterov-accelerated Adaptive Moment Estimation |
| NN | neural network |
| OCR | optical character recognition |
| PANet | Path Aggregation Network |
| Resnet50 | a convolutional neural network architecture |
| ResNeXt-101 | a model introduced in the Aggregated Residual Transformations |
| RetinaNet | is a one-stage object detection model |
| ROI | Region of Interest |
| RPN | Region Proposal Network |

| | |
|---|---|
| R-CNN | region-based convolutional neural network |
| R-FCN | region-based fully convolutional network |
| SGD | stochastic gradient descent |
| SSD | Single Shot Multi-Box Detector |
| ShuffleNet | an extremely efficient CNN |
| SpineNet | learning scale-permuted backbone |
| SqueezeNet | a convolutional neural network |
| TVOD | textual visual object detection |
| VC | video consultation |
| VGA | Video Graphics Array |
| VGG | Visual Geometry Group |
| VGG16 | a convolutional neural network |
| VOD | visual object detection |
| VOTT | Visual Object Tagging Tool |
| VRAM | video RAM, video random access memory |
| YOLO | You Only Look Once |

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
