# Peer review of "Visual Object Detection with DETR to Support Video-Diagnosis Using Conference Tools"

_applsci, doi:10.3390/app12125977_

Round 1
Reviewer 1 Report
The authors should check the ease of access for each acronym description, and also the possibility of reducing the excess of acronyms.
They use the term “Chapter” instead of “Section” to link different parts of the text. They should avoid unnecessary repetition when citing references in the Material and Methods section.
The algorithm logic should be better presented.
The title should not contain the last point (.) as in “2.5.1. Advantages.”.
Figures should be improved since some texts are very small to read.
They should present the number of cases in each class before the pre-processing step, so that the selected procedure may be correctly justified considering the data-hungry nature of the deep learning approach.
They finally stated that "The textual data object detection is feasible having a mean average precision of 0.65." How this result is comparable with other reported approaches to same detection problem? They should analyze different approaches to the same problem in the Introduction and Discussion sections.
Reviewer 2 Report
Some points need to be further clarified:
- The introduction can be written more comprehensively and substantially.
- Unfortunately, some significant work is not considered nor cited, such as:
https://ieeexplore.ieee.org/document/8627998
https://doi.org/10.1016/j.image.2021.116618
https://doi.org/10.1016/j.jflm.2021.102255
https://doi.org/10.1016/j.measurement.2021.110292
- The current experiment results are too insufficient to prove the conclusion given by the authors. The paper also lacks a statistical test to verify the significance of the results.
- There are too few ways to compare. A comparison with existing classical methods is necessary to demonstrate that the authors' results are superior.
Round 2
Reviewer 1 Report
The authors improved the paper.